# Mechanical Consequences of Dynamically Loaded NiTi Wires under Typical Actuator Conditions in Rehabilitation and Neuroscience

**DOI:** 10.3390/jfb12010004

**Published:** 2021-01-08

**Authors:** Umut D. Çakmak, Zoltán Major, Michael Fischlschweiger

**Affiliations:** 1Institute of Polymer Product Engineering, Faculty of Engineering & Natural Sciences, Johannes Kepler University Linz, Altenbergerstrasse 69, 4040 Linz, Austria; zoltan.major@jku.at; 2Chair of Technical Thermodynamics and Energy Efficient Material Treatment, Institute of Energy Process Engineering and Fuel Technology, Clausthal University of Technology, Agricolastrasse 4, 38678 Clausthal-Zellerfeld, Germany

**Keywords:** rehabilitation and neuroscience, shape memory alloy, dynamic thermomechanical material behavior, mechanical loss factor, storage modulus, loading rate- and temperature dependency

## Abstract

In the field of rehabilitation and neuroscience, shape memory alloys play a crucial role as lightweight actuators. Devices are exploiting the shape memory effect by transforming heat into mechanical work. In rehabilitation applications, dynamic loading of the respective device occurs, which in turn influences the mechanical consequences of the phase transforming alloy. Hence in this work, dynamic thermomechanical material behavior of temperature-triggered phase transforming NiTi shape memory alloy (SMA) wires with different chemical compositions and geometries was experimentally investigated. Storage modulus and mechanical loss factor of NiTi alloys at different temperatures and loading frequencies were analyzed under force-controlled conditions. Counterintuitive storage modulus- and loss factor-dependent trends regarding the loading frequency dependency of the mechanical properties on the materials’ composition and geometry were, hence, obtained. It was revealed that loss factors showed a pronounced loading frequency dependency, whereas the storage modulus was not affected. It was shown that force-controlled conditions led to a lower storage modulus than expected. Furthermore, it turned out that a simple empirical relation could capture the characteristic temperature dependency of the storage modulus, which is an important input relation for modeling the rehabilitation device behavior under different dynamic and temperature loading conditions, taking directly into account the material behavior of the shape memory alloy.

## 1. Introduction

To extend the duration of therapeutic sessions, robotic systems are becoming a very efficient tool in the field of rehabilitation and neuroscience [1]. In rehabilitation, robotic tools enable repetitive and repeatable mobilization of the whole limb [1,2] or joint by joint [3]. Managing the motion of effectors by robotic systems gains information of the interconnection between activities of peripheral segments and brain structures, hence this is a promising approach in neuroscience [1,4,5,6,7]. Consequently, there is great interest in portable lightweight devices, where actuation is based on shape memory alloys (SMA). Additionally, SMAs can be embedded in a polymeric matrix to gain actuation capabilities [8,9,10].

The understanding of SMA as a class of metals with the capability of changing their shape due to applied thermal, mechanical or magnetic fields has been studied in the last few decades [7,11,12,13,14,15,16,17]. The physical mechanism behind the shape memory effect or the material’s superelasticity is a displacive solid-to-solid phase transformation where, due to an externally applied field, a spontaneous change of the crystallographic lattice from a low temperature phase “martensite” to a high temperature phase “austenite” and vice versa occurs, hence activating a change of the macroscopic mechanical properties [16,18,19,20,21,22,23]. Specifically, in NiTi alloys, even a so-called “R-phase” is observed which corresponds to a martensite phase, however it is not contributing to the shape memory effect. Further information can be found in Bhattacharya (2003).

Depending on the alloy’s composition, the thermomechanical characteristics can be varied. Nowadays, owing to numerous research efforts, a better understood shape-memory effect of, e.g., Cu–Zn–Al, Cu–Al–Ni and NiTi alloys, allows more sophisticated applications in different fields [24,25,26,27,28]. For medical applications NiTi alloys are typically be used as implants and as superelastic elements and actuators in the field of rehabilitation and neuroscience [1].

However, for NiTi alloys, there are still open issues and unsolved aspects regarding the development of alloys with specific and tailor-made properties for applications concerning temperature and dynamic mechanical loading-dependent material characteristics [29]. These issues have not been fully investigated in the past [29,30]. A large number of commercial products made of SMA are produced in the form of wires [24,31] which further operate as actuators and damping elements in devices. In particular, for a better understanding of the materials’ damping capacity in connection with variations of effective modulus, fundamental knowledge is required [29] to support modelling and prototyping of devices. An important step in the design of a SMA-based rehabilitation actuator is a careful analysis of biomechanical boundary conditions, i.e., resisting loads [1]. This leads to the constraint that force controlled boundary conditions are of significant relevance in rehabilitation actuators, whereas in literature information about the material behavior under these conditions can rarely be found.

Additionally, SMAs are deploying their damping capacity dependent on the external thermomechanical loading. For rehabilitation and neuroscience actuators the shape memory effect is triggered by temperature-induced phase transformation [1] under a respective stress level which is of course below the stress-induced transformation level. Contrary to temperature-induced phase changes, stress-triggered phase transformations in SMAs exhibit higher damping capacities and consequently they have been more focused on in materials science in the past with respect to dynamic amplitude and frequency effects [32,33]. The majority of experimental studies are performed under displacement/strain controlled loadings rather than force/stress controlled. However, in neuroscience and rehabilitation applications, experiments under force/stress controlled loadings are needed and are of particular interest for product engineering.

Therefore, in this work the focus lies on the understanding of dynamic mechanical properties of NiTi-SMA wires, under controlled temperature and force fields for various mechanical loading frequencies in connection with effective modulus change, especially in the temperature-triggered transformation zone. Earlier experimental investigations on NiTi alloys revealed properties influencing the materials’ damping capacity in the transformation zone [34]. It was shown that material softening takes place during the transformation referred to the observation that the modulus of the SMA was lower in the transition region than the moduli of austenite and martensite in the pure phase state. The softening effect could be also explained and predicted theoretically by using statistical mechanics-based modeling strategies developed by Oberaigner and Fischlschweiger [15]. Furthermore, Yastrebov et al. [16] modeled this phenomenon with the Lattice–Monte Carlo approach. In addition, Wang and Sehitoglu [35] highlighted the dramatic difference of the macroscopic modulus influenced by the multivariant state of martensite. The selection of martensitic variants in particular is dependent on external applied stress fields and is sensitive to tension and compression. Hence, moduli of NiTi alloys are different in tension and in compression [35]. Here, we analyze the macroscopic behavior of NiTi alloys under dynamic thermomechanical loading under force controlled conditions to gain insights into the loading-dependent damping and modulus behavior of SMA wires with different geometries and materials’ compositions. This information is relevant for selecting NiTi-wires as actuator and damping elements in rehabilitation and neuroscience devices. We present the experimental methodology to characterize the dynamic thermomechanical behavior of NiTi alloys. Special emphasis is given to the analysis of the stiffness and the mechanical transient properties, i.e., loss factor. Furthermore, we seek to find a simple empirical relation for the temperature-dependent effective modulus behavior. This is particularly important for taking into account the respective SMA material properties in modeling rehabilitation and neuroscience devices.

## 2. Materials and Methods

### 2.1. Materials

Table 1 lists the compositions, the diameters of the round wires, and the surface conditions of the alloys. All wires were straight annealed (superelastic) by the manufacturer and, with the exception of alloy M5, the materials’ surfaces were oxidized. Quasistatic tensile tests (Electromechanical actuator, TestBench, TA, ElectroForce, New Castle, DE, USA) at room temperature (22 °C) were performed in order to determine the Young’s moduli of the alloys studied. The moduli are also listed in Table 1. Due to the different surface treatments, specimens are in principle also subjected to different heat-treatment steps. In addition, the mechanical polishing step can also alter work hardening in the original wire specimen. Microstructural variations due to heat treatments lead in turn to significant variations of macroscopic mechanical properties, e.g., modulus [36,37]. This explains also the variation of the modulus behavior found in this study (cf., Table 1).

All alloys show R-phase transformation behavior and transform from B2 cubic austenite sequentially via rhombohedral R-Phase to B19′ monoclinic martensite. Moreover, functional fatigue properties it is typically for alloys with R-phase transformation behavior that small transformation strains are achieved [38].

The alloys were selected according to their practical use as wires with differences in alloy composition (stiffness), geometry and surface finishing, in order to analyze the respective behavior under dynamic thermomechanical loadings. As a remark, oxidized surface finishing is particularly relevant in cases where the device has to fulfill sterilization requirements.

### 2.2. Characterization Methodology

In principle, there exist different experimental methodologies for studying the modulus behavior of NiTi alloys. This spans from typical quasistatic mechanical tests, ultrasound methods up to thermomechanical and dynamic thermomechanical methods (e.g., [38,39,40,41,42]). It is already known that pure phase elastic properties of NiTi alloys with R-phase transformation obtained by ultrasonic transmission measurements agree well with those obtained by mechanical tests. However, a disagreement appears in the R-phase transformation region [38]. In ultrasonic measurements elastic constants are determined by evaluating the propagation speed of longitudinal and transversal ultrasonic waves. As it is documented in the work by Sittner et al. [38], the wave speed is additionally affected by microstructure, which results in questionable experimental results in the temperature ranges, where phase mixtures exist. In this work the focus is on frequency dependent dynamic modulus behavior especially within the transformation zone of NiTi alloys showing R-phase transformation behavior under force controlled conditions.

Here, the characterization methods included differential scanning calorimetry (DSC) for determining the character of phase transformation and dynamic thermomechanical analysis (DTMA) under tensile loading conditions. DSC is further used to investigate the impact of thermal rate effects on shifting the phase transformation behavior with respect to heating and cooling. Therefore, the impact of occurring thermal rate changes in the temperature control chamber of DTMA on phase transformation for ensuring a correct interpretation of DTMA measurements. An equivalent DSC experimental procedure—as reported by Zheng et al. [11]—was conducted using a Mettler Toledo DSC 822 with a DDK FRS5 sensor, an intercooler TC100MT-NR and a gas controller GC 200. The heating and cooling rate was 10 K/min for the standard measurements. This corresponds to the typical heating rate in the temperature control chamber. Herein, a heating/cooling rate-dependent DSC measurement was performed for one of the alloys investigated to analyze the kinetic effect regarding the influence on the transformation temperatures. Frequency sweeps were performed under isothermal conditions, however, rates for setting the desired temperature can be varied and may influence the specimen’s thermomechanical behavior. In order to determine the impact of thermal rate effects on the thermomechanical responsiveness of the material, heating and cooling rates of 0.1 K/min and 1 K/min were additionally examined.

DTMA (TestBench, TA, ElectroForce, New Castle, DE, USA) was used to study the frequency dependence of modulus behavior of the samples given in Table 1, working out especially the discrepancy of storage and loss factor for the respective sample. During the DTMA experiments, a force-controlled sinusoidal excitation was applied under isothermal conditions. A detailed description of the measurement setup and procedure can be found in the work by Cakmak and Major [43]. The excitation mean level was 25 MPa, and the dynamic amplitude was 5 MPa during the frequency sweep from 1 Hz to 21.5 Hz with six intervals per logarithmic decade at temperatures changing in 10 K (5 K) increments from 353.15 K to 243.15 K. The heating and cooling rate between the isothermal steps was in a range of (10 ± 1) K/min. At each temperature step, the initial length at thermal equilibrium was considered. The storage modulus E’ and the loss factor tanδ = E″/E’ were determined from each isothermal measurement.

## 3. Results

### 3.1. Differential Scanning Calorimetry (DSC)

Table 2 summarizes the thermal characteristic transition temperatures from DSC measurements. All alloys revealed an R-phase and the corresponding start and finish transition temperatures are listed in Table 2.

The kinetic effect during the DSC measurement was investigated for alloy M5 (see Figure 1). With an increasing heating rate, the austenite transition temperatures (AS and AF) also increased slightly. However, R-phase and martensite transition temperatures decreased slightly with a higher cooling rate. Generally, the kinetic effects on the transformation temperatures were relatively low and in accordance with the conclusion of Nurveren et al. [44]. It can be concluded that the heating/cooling rate for setting the required temperature for DTMA experiments will not affect substantially the behavior of the materials under investigation. Therefore, temperature rate effects in the DTMA with the current parameters can be excluded, this holds true for peak, as well as for start and finish temperatures.

### 3.2. Dynamic Thermomechanical Analysis (DTMA)

In the following diagram (see Figure 2), the examined E’ results at the temperatures investigated are illustrated for all NiTi alloys. First of all, the experimental data points indicate that the materials’ storage modulus is, as earlier discussed and expected, strongly temperature-dependent. Generally, the course of the data points is similar to earlier reported results in the literature from model predictions and experimental findings [15,16,30,34,43,45] and is characteristic for NiTi alloys with different chemical compositions. The temperature determines the actual state of the microstructure regarding the content of austenite, martensite and the intermediate phase (R-phase).

Some of the alloys reveal a very narrow transition region, as is the case for alloys M1, M3 and M5. Alloy M4 shows only a part of the transition region within the investigated temperature range. This is due to the alloy’s formulation with transition temperatures higher than those of the others. Alloys M2 and M5 are basically the same formulation, but with oxide and oxide-free surfaces, respectively. The comparison of M2 with M5 reveals that the temperature-dependent storage modulus characteristic is also influenced by the surface treatment. M2 exhibits, generally, higher moduli values and, between 260 K and 320 K, a rather constant modulus (~28 GPa). In contrast, the transition region of alloy M5 starts at 320 K and ends at 313 K. Comparing M5 with M2, the storage modulus within the transformation region of M5, where phase mixture exist, is in contrast to M2 not constant, rather it changes continuously. This information is of particular interest, because it shows, that an oxide layer on the NiTi wire increases the dynamics storage modulus with the factor 2 and is hence a remarkable parameter for controlling the dynamic stiffness of NiTi wires and consequently of the actuator device. Due to the oxide layer a more constant modulus behavior within the transformation zone exists in a broader temperature interval, this has a high impact on material’s selection for the respective use case.

In addition, the dynamic storage Young’s moduli at room temperature are lower than those of the quasistatic tensile tests (cf., Table 1). This is in analogy to earlier investigations [46,47], where cyclic softening was observed and analyzed.

Moreover, from the DTMA measurements, the mechanical loss factor tanδ was evaluated, and the temperature- and excitation frequency-dependent characteristics of the alloys can be seen in Figure 3. To facilitate the observability of the temperature-dependent loss-factor trend, the average tan δ over the examined frequency range is shown in the diagram. In addition, the maximum and minimum tan δ (dashed) lines are illustrated, and arrows indicate the observed tan δ within the investigated frequency range. Generally, the evaluated tan δ values are reasonable and are within the limit of the mechanical loss factors of steel (0.002 to 0.01) (cf., [48]). A counterintuitive frequency dependency is observable by comparing the mechanical loss factor with the earlier presented storage moduli data (see Figure 2). The moduli of the alloys are clearly frequency independent, while the loss factors alter in discrepancy with the former with varying excitation frequencies. The lower the loading frequency, the higher the loss factor; whereas the course of the temperature-dependent curve is equivalent. This discrepancy between the mechanical transient and storage behaviors of NiTi alloys affects the performance of devices during dynamic loading. It mainly affects the damping behavior, while the resonance frequency of the structure is unaffected. On the other hand, the difference between the investigated alloys’ tan δ characteristics was not as distinctive as the modulus characteristics; less variation between the alloys was observed. Alloys M3 and M5 revealed a similar characteristic progression and showed an increase from low temperatures up to a peak followed by a decline to a minimum; whereas the other alloys were, in principle, constant between 270 K and 320 K.

The discrepancy of frequency dependence of storage modulus and loss factor can be physically explained, by the fact, that the viscos part of all NiTi alloys investigated in this study is almost negligible and therefore a frequency independent storage modulus behavior was observable. The modulus behavior itself is, however, strongly dependent on respective microstructure realized by different chemical compositions and by oxide surface layers of the wires. The measured storage modulus behavior is lower than the typically obtained modulus from quasistatic tests and ultrasonic measurements. The reason lies in the dynamic measurement procedure carried out under controlled stress and not strain boundary conditions. Hence, internal strains are developing due to the cyclic loading to higher values ending in a plateau by constant external force conditions. Consequently, lower storage moduli are obtained with respect to for example quasistatic measurement conditions. The frequency-dependent loss factors occur due to internal friction phenomena, strongly pronounced especially in the phase mixtures regions. Of particular interest, is that internal friction seems to be similar within the investigated NiTi alloy ending in a similar, however, frequency-dependent, damping capacity of the alloys.

Preparing the experimental storage modulus differences according to chemical composition and surface treatment for designing actuator elements and taking them into account in model calculation, in the next paragraph an empirical description of storage modulus behavior is presented.

From the storage modulus curves, it can be figured out that the modulus behavior is temperature T dependent and frequency f independent, and that E (T, f) = E (T). E (T) can be modeled by the biphasic function type: (1)$$E(T)=E_{\mathrm{PT}}+\frac{E_{A}-E_{\mathrm{PT}}}{1+e^{\left(T-T_{0_{1}}\right)h_{1}}}+\frac{E_{M}-E_{\mathrm{PT}}}{1+e^{\left(T_{0_{2}}-T\right)h_{2}}}$$
where E_PT_ is the soft modulus point, meaning the modulus point with the lowest value in the transformation zone, E_A_ is the modulus of the austenite dominant state of the alloy, E_M_ is the respective modulus of the martensite dominant state, the temperature, T_01_ and T_02_ are the temperatures related to the inflection points of the curves, and h_1_ and h_2_ are dimensionless constants of the exponential functions. For the alloy M5, the modeled curves are illustrated in Figure 4. At each measured excitation frequency, the parameters for the function in Equation (1) were determined. In Figure 5, the moduli parameters of the biphasic function are shown. Up to 21.5 Hz, the determined parameters are almost constant, and this confirms that the moduli of the alloys are frequency-independent.

The capability of the proposed model (Equation (1)) is shown in Figure 2. The DTMA data are sufficiently adjusted by the biphasic function, indicating the applicability of the function. From the experimental data the model parameters determined are shown in Table 3. The difference of the alloy composition (M1 to M5) leads also to a variation of the mechanical behavior, in which the moduli related to the phase transition E_PT_ (R-phase) are lower than those of the other phases (E_A_ and E_M_). All model parameters (see Table 3) are constant in terms of excitation frequency. The proposed mathematical function is capable of describing the different temperature-dependent modulus characteristics of the alloys and justifies the applicability to NiTi alloys. Phase transition temperatures (T_01_ and T_02_) are captured by the model in Equation (1). In order to describe the change in the moduli during phase transitions, the exponential coefficients h_1_ and h_2_ are defined. Therefore, these coefficients are composition-dependent and account for the behavior of the moduli changes (cf., M2 vs. M5 in Figure 2).

## 4. Summary and Conclusions

A series of dynamic thermomechanical analyses (DTMA) were conducted to study the frequency dependency of the mechanical behavior of NiTi alloys with different chemical compositions and wire dimensions. The phase transformation is induced thermally, where the applied dynamic stresses are in a range below the level of stress-induced transformation. This setup becomes highly relevant for the understanding of SMA wires which are essential actuator elements in the field of rehabilitation and neuroscience. The experiments were performed force controlled, taking into account practical actuator conditions from rehabilitation and neuroscience application. Based on the aforementioned conditions, the storage modulus and the mechanical loss factor were determined and analyzed over various frequencies for typical temperature courses. All alloy compositions and geometries revealed an almost frequency-independent storage modulus, while the mechanical loss factor showed a pronounced dependency on the excitation frequency. This discrepancy is of particular interest when dynamically loaded devices are considered. The damping capability is altered by the external mechanical loading excitation and has consequences in the overall performance of the material. At the same time, the stiffness of the material remains unchanged and hence the resonance frequency of the structure. Furthermore, it is shown that the dynamic modulus for force-controlled loading conditions, as is the case for the application, is considerably lower than the quasistatic measured modulus. This should be taken into account in the design of generally dynamically loaded rehabilitation actuators which are bound to controlled force conditions. To exclude any kinetic effect during the temperature ramp in DTMA between the temperatures investigated and during the frequency sweep protocol, differential scanning calorimetry measurements were performed for different cooling rates and hence thermal properties of several alloy compositions were characterized.

It can be concluded that the NiTi alloys under investigation show almost no kinetic effect to heating/cooling rate; however, they exhibit kinetic effects to mechanical loading rate (frequency). Only the damping behavior (tan δ) is affected by the mechanical rate; the stiffness (E’) is frequency independent.

Additionally, a simple empirical function describing the temperature-dependent storage modulus behavior is proposed. This enables temperature-dependent dynamic storage modulus behavior to be considered directly for the selected materials as a relevant material input property for structural actuator design and modelling.

## Figures and Tables

**Figure 1 jfb-12-00004-f001:**
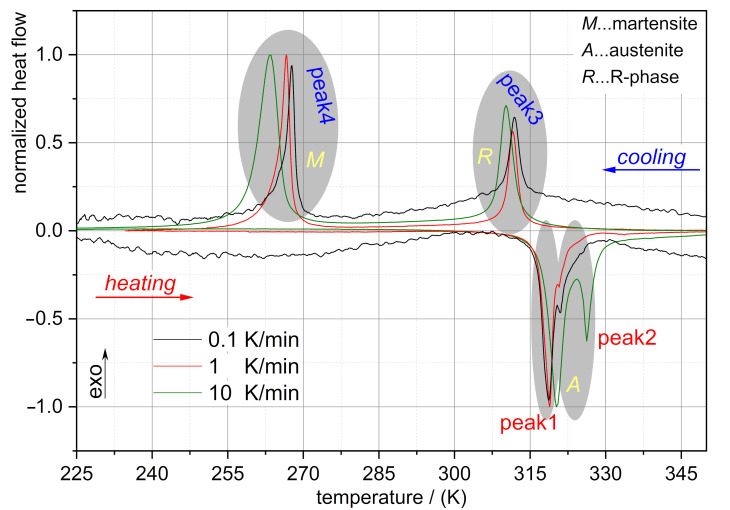
Differential scanning calorimetry (DSC) results of the M5 alloy showing the temperature memory effect for three different heating/cooling rates. For the sake of visualization and comparability the heat flow is normalized.

**Figure 2 jfb-12-00004-f002:**
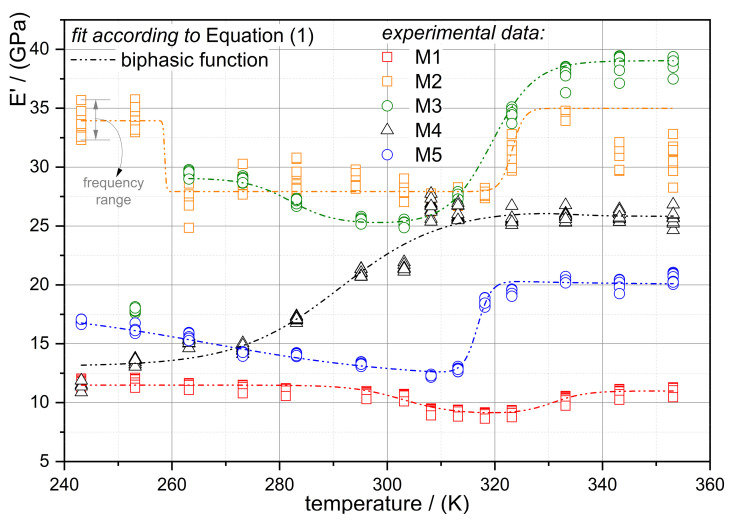
Comparison of the temperature dependent moduli of all alloys within the examined frequency range of 1 to 21.5 Hz. Dashed lines are the modeled curves according to Equation (1).

**Figure 3 jfb-12-00004-f003:**
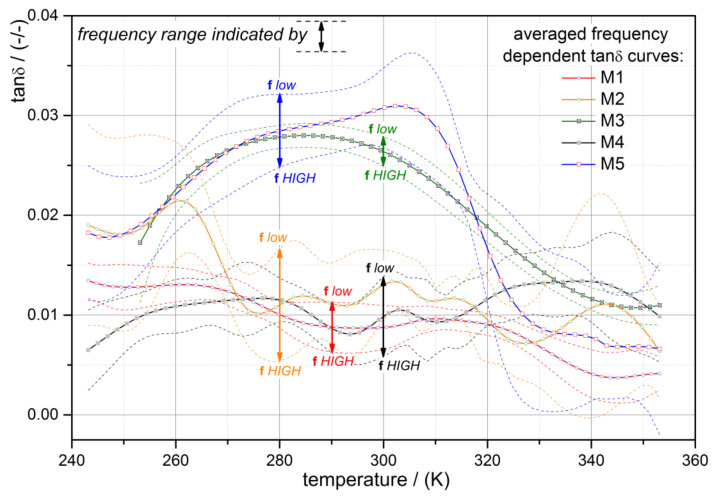
Comparison of the temperature dependent mechanical loss factor tan δ determined from dynamic thermomechanical analysis (DTMA) experiments). Full lines represent the over the frequency range averaged curves and the dashed lines show the respective high- and low-frequency lines.

**Figure 4 jfb-12-00004-f004:**
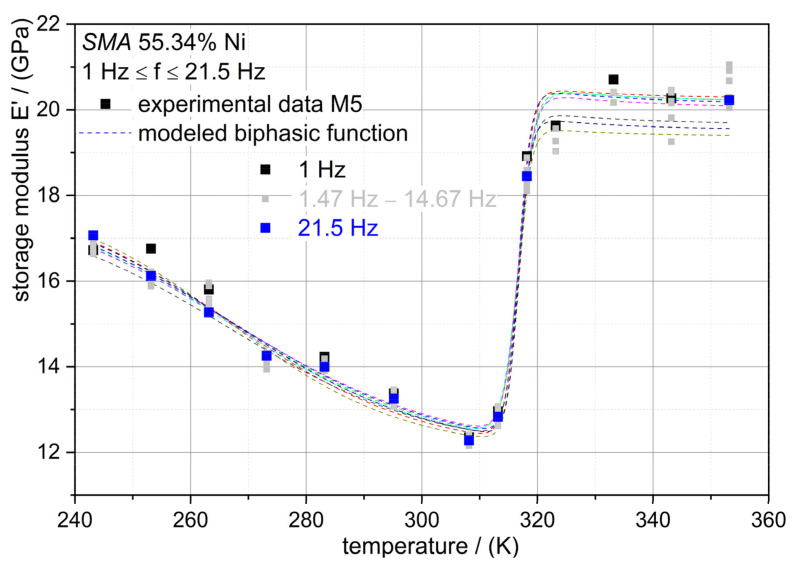
DTMA results of the alloy M5 showing the temperature-dependent storage moduli. Data points represent the experimental data in the investigated frequency range; dashed lines are the fitted curves of the data according to Equation (1).

**Figure 5 jfb-12-00004-f005:**
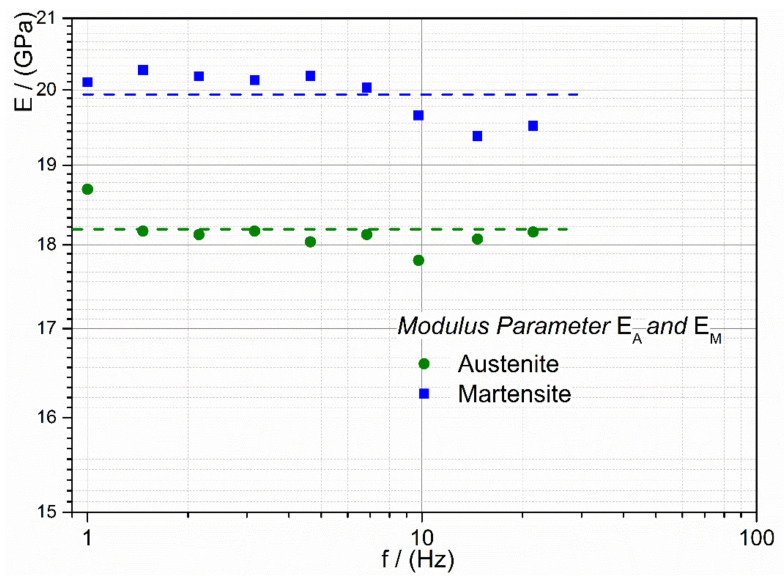
The frequency-dependent modulus parameter of the biphasic function (Equation (1)) for the alloy M5.

**Table 1 jfb-12-00004-t001:** Specifications of the NiTi alloys.

Alloy	Ni	Ti	Diameter	Surface Finish	Young’s Modulus E at 22 °C
	%	%	µm		GPa
M1	55.60	44.40	1000	Oxide	13
M2	55.34	44.66	320		30
M3	55.77	44.23	323		33
M4	54.80	45.20	305		24
M5	55.34	44.66	760	Oxide free	15

**Table 2 jfb-12-00004-t002:** Thermal characteristics of the investigated alloys. Austenite start and finish transition temperatures (A_S_ and A_F_), R (rhombohedral or R-Phase) start and finish temperatures (R_S_ and R_F_) and martensite start and finish temperatures (M_S_ and M_F_).

Alloy	T-Rate (K/min) Heating/Cooling	A_S_ (K)	A_F_ (K)	R_S_ (K)	R_F_ (K)	M_S_ (K)	M_F_ (K)
M1	10	327.86	339.24	329.84	309.89	272.86	215.08
M2	10	324.30	341.10	327.73	301.16	298.53	245.48
M3	10	290.99	326.26	323.91	283.45	-	-
M4	10	355.61	362.82	336.65	332.19	319.28	309.92
M5–10	10	317.19	328.46	313.30	307.58	266.61	258.49
M5–1	1.0	316.24	323.15	313.71	309.84	267.82	264.33
M5–0.1	0.1	316.12	323.05	314.43	309.54	268.89	265.93

**Table 3 jfb-12-00004-t003:** Parameters for the proposed model in Equation (1) for all investigated Ni–Ti alloys.

Alloy	E_PT_ (MPa)	E_A_ (MPa)	E_M_ (MPa)	T_01_ (K)	T_02_ (K)	h_1_	h_2_
M1	9025	11492	10994	302.93	330.83	0.21	0.32
M2	27922	33941	34996	258.66	323.22	7.01	0.90
M3	25152	29070	39038	282.05	319.44	0.24	0.25
M4	12500	13098	25850	333.82	291.28	0.24	0.11
M5	12020	17810	19657	266.55	315.75	0.06	0.87

## Data Availability

The data presented in this study are available on request from the corresponding author.

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
