# Peer review of "Mechanical Consequences of Dynamically Loaded NiTi Wires under Typical Actuator Conditions in Rehabilitation and Neuroscience"

_jfb, 2021, doi:10.3390/jfb12010004_

Round 1

Reviewer 1 Report

The paper describes mechanical consequences of dynamically loaded NiTi wires under typical actuator conditions in rehabilitation and neuroscience.

It's an interesting work, the abstract is clear and presents very well the study, but the introduction is limited from the point of view of the most recent bibliographic analysis, the most recent references are from 2015, it may be useful to understand the most recent evolution of SMA actuators.

The methods are adequately presented, however, specifying in more detail the frequencies chosen for DTMA rather than just the range or inserting the oxide thickness could be useful.

The experimental results are adequately supported, as far as the empirical model is concerned, the exponents h1 and h2 which depend on the type of regression for composition M2 are very different in particular h1, and this is not commented on by the authors. 

Author Response

Please see our reply in the attached word file.

Reviewer 2 Report

The paper describes the behaviour of NiTi shape memory alloy under dynamic loading.

The Indroduction is detailed in terms of the discussed applications.

The description of the shape memory behaviour is very vague, completely avoiding the definition of the terms "martensite", "austenite" and "R-phase", which are used in the Results and discussion part.

The prime novelty of the paper is not stated clearly. There is a lot of papers describing the dynamic loading response of nitinol so far.

The research design is well-elaborated, but it is not clear why there is a different Young's modulus in tzhe case of M2 and M5 samples, which have the same chemical composition.

In Results and Discussion, the sensitivity of some phase transformations to the heating rate is described, but not explained. Why is it so?

The Summary and Conclusions part is not fully inline with the Results and Discussion, because it states that there is almost no effect of the heating rate on the phase transformations, even though the evident dependence is desribed in Results and Discussion.

Author Response

Please find attached our reply to the reviewer's comments.

Round 2

Reviewer 1 Report

After review the references were updated and some more useful information was added; there is still some gaps in the explanations, but it could be published as is.

Author Response

Thank you very much for your feedback. With our latest revision, we added some information to enhanced the quality of our paper.

Reviewer 2 Report

The paper has been improved significantly and most of the comments have been taken into the account. However, one point is still controversal:

Is it really possible that a thin surface layer of the oxide affects the bulk Young's modulus so dramatically (two times difference)? The values presented in Table 1 are strange in general and have to be verified.

Author Response

Thanks again for your input. Now, we believe to have understood our concern. We reviewed also Young's moduli and also the storage moduli E' data, however the results are as presented. The effects on the Young's modulus are addressed starting from line 112 to 116:

"Due to the different surface treatments, specimens are in principle also subjected to different heat-treatment steps. In addition, the mechanical polishing step can also alter work hardening in the original wire specimen..."